# The Rise of Retinal Organoids for Vision Research

**DOI:** 10.3390/ijms21228484

**Published:** 2020-11-11

**Authors:** Kritika Sharma, Tim U. Krohne, Volker Busskamp

**Affiliations:** 1Department of Ophthalmology, Faculty of Medicine, University of Bonn, 53127 Bonn, Germany; Kritika.Sharma@ukbonn.de; 2Department of Ophthalmology, Faculty of Medicine and University Hospital Cologne, University of Cologne, 50937 Cologne, Germany; tim.krohne@uk-koeln.de; 3Center for Regenerative Therapies (CRTD), Technical University Dresden, 01307 Dresden, Germany

**Keywords:** human retina, PSCs, 3D retinal organoids, retinal neurodegeneration, disease modeling

## Abstract

Retinal degenerative diseases lead to irreversible blindness. Decades of research into the cellular and molecular mechanisms of retinal diseases, using either animal models or human cell-derived 2D systems, facilitated the development of several therapeutic interventions. Recently, human stem cell-derived 3D retinal organoids have been developed. These self-organizing 3D organ systems have shown to recapitulate the in vivo human retinogenesis resulting in morphological and functionally similar retinal cell types in vitro. In less than a decade, retinal organoids have assisted in modeling several retinal diseases that were rather difficult to mimic in rodent models. Retinal organoids are also considered as a photoreceptor source for cell transplantation therapies to counteract blindness. Here, we highlight the development and field’s improvements of retinal organoids and discuss their application aspects as human disease models, pharmaceutical testbeds, and cell sources for transplantations.

## 1. Introduction

Vision is our most dominant sense, and by far the most complex and highly developed [1]. Most of our experiences of the world, and our memories of it, are based on this perceptual modality [2]. Image perception begins with the detection and processing of light, wherein a set number of photons focused by the cornea and lens enter the proximal surface of the retina. The retina is the innermost neuronal region at the back of the eye: it facilitates the conversion of light into electrochemical signals, processes images, and transmits this visual information to higher brain areas via the optic nerve. The retina consists of different neuronal and glial cell types, such as the light-sensitive rod and cone photoreceptors (PRs) that catch photons via their photopigments and mediate the retinal function described above. Further, these photons are readily converted into electrical signals through a complex signaling cascade. These signals are then further processed by inner retinal neurons and relayed through the optic nerve to several areas of the brain, mainly the primary visual cortex in the occipital lobe [3]. After processing, this information becomes associated with any previous innate or learned memories, and is stored for future reference [1].

Impairment at different steps of image or information processing can lead to different eye disorders, graded from restorable to non-correctable unilateral or bilateral vision loss, thereby significantly reducing the quality of life [4]. As life spans increase, and the global population rises exponentially, these eye disorders are causing increasing problems. Age-related macular degeneration (AMD) is the most common cause of blindness in the western world [5,6,7] and is expected to affect 388 million people globally by 2040 [8]. Retinitis pigmentosa (RP) is the leading cause of inherited blindness, showing a prevalence of 1 in 4000, with over 70 genes responsible for vision loss [9]. RP results in degradation of rod PRs, followed by the loss of cone function, resulting in complete blindness. Significant rod degeneration has already occurred before the patient exhibits the first symptoms. There are significant therapeutic advances for treating the neovascular subtype of AMD, such as intravitreally administered anti-vascular endothelial growth factor substances [10]; however, there are currently no effective, targeted and efficient treatment options for retinal cell degeneration in the atrophic subtype of AMD and in RP [11]. In recent years, considerable progress has been made in understanding these retinal diseases and in identifying potential therapeutic targets for intervention [12]. Most of this research is based on animal models as human *post-mortem* retinal tissue is extremely fragile [13], not expandable, shows donor and preparation-dependent batch effects and is often unavailable. Stem cells offer new opportunities for modelling retinal diseases, and stem cell-derived 3D retinal organoids have recently become the focus for therapeutics and disease research. Here we summarize these developments in 3D retinal organoid technology for disease modeling and personalized medicine.

### Examining the Retina In Vitro

Investigating the underlying morphological, physiological, and functional aspects of a pathology is a prerequisite for finding a cure. Many retinopathies have been illustrated by imaging surgically excised retinal tissue. However, the survival of human retinal tissue in vitro depends on fast isolation and a continuous oxygen supply to keep the tissue electrically functional [14]. In addition, there is extremely limited availability of donated human retinal tissues which have specific retinal pathologies and their progression states. The disease states of donated *post-mortem* retinal tissues are unclear at the time of extraction, and the tissues come mostly from healthy donors. Therefore, widespread use of human retinal tissues as functional in vitro testbeds for vision research is impractical. Hence, researchers have examined the next best option, animal retinas with retinal pathologies secondary to genetic or interventional modifications. Even though animal models have expanded our understanding of developmental, physiological, functional, and regenerative attributes of the retina, there are noticeable disadvantages, primarily challenges in extrapolating the findings of rodents to humans due to a different physiology such as absence of a macula and differences in color vision. Moreover, there are ethical concerns when using non-human primate models. One solution would be to generate functional retinal tissues in vitro from an unlimited and ethically acceptable human cell source.

Pluripotent stem cells represent such an unlimited cell source. Mouse embryonic stem cells (ESCs) were discovered in 1981 [15] and human ESCs in 1998 [16]. Human ESCs were established as efficient, reproducible, and easy to handle human-based cellular models for disease investigation, as well as for drug testing. However, due to ethical concerns resulting from the disruption of embryos to generate ESCs, the groundbreaking discovery of induced pluripotent stem cells (iPSCs) in 2006 (mouse) [17] and 2007 (human) [18] overcame these ethical roadblocks and boosted biomedical stem-cell research. These autologous cells can be generated from the patient’s own cells and thus avoid immune rejection caused by the human leukocyte antigen (HLA) [19], and have encouraged an era of personalized cellular medicine following transplantations of stem cell-derived somatic cells or tissues. Currently, banks of iPSC cell lines are being evaluated for clinical application: these are derived from homozygous HLA donors with immunological compatibility for a larger population set, also called the super donors. These banks are called haplobanks and they have great potential for broadening the horizons for therapeutics and regenerative medicine. Like all pluripotent stem cells, iPSCs are able to form all cell types of cell lineages including retinal cells when incubated with appropriate fate-specification factors in an in vitro cell culture environment [20]. These infinitely expanding, reprogrammed pluripotent stem cell-derived 2D cultures have established their footing in unfolding complex developmental pathways; and have entered the field of drug testing and discovery, disease modelling, and even cell replacement therapies to functionally and physiologically compensate for lost or degenerated cells [21,22,23,24,25,26,27,28]. These stem-cell-derived 2D cultures, however, fail to entirely replicate the structural, physiological and functional aspects of the retina. To reinstate cell-cell interactions and the natural course of cell signaling-assisted development as seen in retinogenesis in vivo, 3D organoids offer a stable, efficient model for analysis of retinal pathologies.

## 2. The Rise of Retinal Organoids–Technical Development and Advances

First efforts to generate 3D retinal tissues in vitro began by promoting and enhancing cell-cell interactions that mimicked the 3D in vivo environment. This laid the foundation for 3D organ-like structures, or so-called organoids. Organoids are autologous tissues derived from pluripotent stem cells (PSCs) in vitro, either via self-organization or guided by a scaffold. Therefore, they structurally and functionally mimic the in vivo environment. From Wilson’s sponge aggregation experiment in 1907 [29] to today’s complex, functional in vitro architecture, research in organoids has come a long way. The 1980s was the decade when human organoids were determined [30] and, ever since, various groups have extended their expertise in generating different organoids representing numerous mini-organs in the lab environment. These have included intestinal, prostate, brain and retinal organoids from embryonic stem cells (ESCs), adult stem cells (ASCs) and from induced pluripotent stem cells (iPSCs) [31,32,33,34,35,36,37].

The era of retinal organoids started with serum-free and modulating factor-free differentiation of mouse ESCs [38] and human ESCs [37] that generated eye-field precursors in 3D retinal clusters, expressing cone-rod homeoboxes (CRX) and opsins. However, stratified cell-specific layered organization was first generated from mESCs in 2011 [38]. These free-floating, serum-free aggregates of embryoid bodies with matrix components from Matrigel generated prominent optic-cup vesicles after undergoing a series of evagination and subsequent invagination steps. These vesicles underwent morphogenesis after self-patterning, self-directed organization, and stepwise domain-specific regulation of cellular architecture, like in vivo retinogenesis. Soon this retinal organoid protocol was adapted for hESCs by adding extrinsic factors such as temporally-regulated antagonists for Wnt, sonic hedgehog, and fibroblast growth factor [38,39]. These retinal organoids successfully exhibited stratified neural retina (Figure 1), generating nascent chemical and electrical retinal synapses [40]. Many different protocols have been developed based on the initial findings: they use different exogenous factors and long-term high-oxygen culturing conditions. To increase the quantity of organoids formed per batch, a protocol was developed which applied trisectioning to the forming organoids. This led to the efficient generation of large stratified retinal organoids, not requiring any evagination of optic-vesicle-like structures. Thereby, the overall quantity of retinal organoids, including the number of PRs, was significantly increased [41]. In 2014, the formation of outer segments (the light-sensitive subcellular components of PRs) was shown in human-cell-derived 3D retinal organoids. Functional light responses were recorded from 27-week-old retinal organoids [42] for the first time. These morphologically functional organoids expressed markers for the phototransduction protein (recoverin) and the synaptic vesicle protein found in rods. This probably occurred because the organoids survived for longer, enabling PRs to mature both morphologically and topologically, and thus eliciting a downstream functional output. It should be noted, however, that these functional events were relatively rare and occurred only in 2 out of 13 randomly selected PRs.

To prolong the survival of these organoids, culture conditions were enhanced using a pre-defined bioreactor that significantly improved the laminar stratification of the retinal layers, generating a high number of mature PRs with clearly visible cilia and nascent OS-like structures with stacked membranous disks [43,44]. The yield of PRs was also significantly improved, with 1 million cells extracted from 120 organoids, enough to transplant into mice [44]. However, the lack of a vascular system within the organoids often led to inefficient oxygen and nutrient supply, which to some extent also explains the limited size and high level of necrosis [45,46]. Impressive new approaches include improving the yield of organoids using hydrogels [47], and speeding up maturation of PRs with intact retinal ganglion cells (RGCs) by supplementing with IGF1 [48] By enhanced vascular-like perfusion, the morphological maturation of PRs and their interactions with co-cultured retinal pigmented epithelium (RPE) improved. Postsynaptic density 95 and c-terminal binding protein antibodies identified the evenly distributed PRs in the outer retina at days (D120-D160). This photoreceptor development corresponds to in vivo human retinogenesis. Photoreceptor outer segments (POS) structures emerge in vivo at around week (W23-W25), and POS structures established in these organoids appeared between W18 and W28, a remarkably similar developmental pattern [49]. Appearance of cilia emerging from mitochondria which inner segments and cilia emerging from them present around W13 and outer segments appeared around W21 similar to previously assessed protocols [50,51]. Many improvements in generating retinal organoids from iPSCs have demonstrated that the addition of retinoic acid and taurine between D90 and D120 of differentiation enhances the formation of rod and S-cone PRs [52]. Very recently, an improved 3D retinal organoid protocol was published, which demonstrated a proper stratification of all retinal layers, including light-sensitive PRs that were functionally connected to inner retinal neurons [14]. Importantly, this protocol also highlighted differences in human iPSC lines regarding their competence in generating layered retinal tissues. These cell-line-specific differences are an important parameter to be considered when establishing different protocols for retinal organoid production. Normally, every research lab uses its own default set of lines, which might be suboptimal for protocols from other laboratories.

### 2.1. Retinal Organoids for Disease Modeling

Connecting cilia are important subcellular structures within the light-sensitive outer segments of photoreceptors. Ciliary function has been studied extensively, from its biogenesis to its detrimental dysfunction in rods and cones, which can cause blindness. The underlying pathological mechanisms are being investigated [53]. Patient-derived human iPSCs, including retinal organoids, offer a new option for uncovering the underlying mechanism of the disease (Figure 1). For example, the mutation of *CEP290*, a primary ciliary protein, was studied in patient-derived retinal organoid structures [54]. Retinal organoids derived from *CEP290*-mutated LCA (Leber congenital amaurosis) and JSRD (Joubert syndrome and related disorders) were examined to understand these ciliopathies. The results corroborate the previously proposed function of CEP290 in gating-specific ciliary proteins, impacting the biogenesis and transport of cell types [55]. Disease-specific retinal dysfunction, and its dysregulated molecular counterpart, mirrored an early-onset retinal degeneration phenotype that had previously been discovered clinically [54]. Recently, a late-onset RP model was also established in an in vitro 3D organoid system underlying the *PDE6B* mutated phenotype [56]. Another project developed retinitis pigmentosa GTPase regulator (*RPGR*) mutation-specific RP retinal organoids that assist in examining the molecular dysregulation of this specific phenotype. It was shown that the *RPGR* mutation resulted in defective PR morphology and localization, shortened cilia, an altered transcriptional profile, and even dysregulated electrophysiological output when compared to a healthy control. As a proof of concept, CRISPR-Cas9 (summarized in Figure 1C) genomic engineering was applied to repair the *RPGR* mutation, which led to a significant improvement [57]. A similar study has been established with an X-linked juvenile retinoschisis retinal organoid model [58]. Ciliary F-actin assembly was recently studied in retinal organoids, concluding that the role of the PR cilium actin regulator (PCARE) and Arp2/3 complex activator is to regulate the formation of primary cilia that drive disc formation in the outer segments (OS) of functional PRs. As a proof of concept, pharmacological inhibition of actin polymerization has also been shown to mirror the *PCARE* mutation phenotype, and that of *PCARE-/-* mice [59].

Retinal organoids are essential for exploring highly heterogeneous diseases such as RP. One of several RP mutations is the hypomorphic mutation in the tRNA nucleotidyl transferase CCA adding 1 (*TRNT1*) gene, which severely affects the PRs and causes early-onset RP. To replicate the disease in its patient-specific genotype, patient iPSCs were used to generate retinal organoids. These organoids displayed similar TRNT1 protein augmentation, deficits in autophagy, and general pathophysiology of cell types that are otherwise inaccessible in living patients [60]. Similarly, retinal organoids were also derived from a patient-specific iPSC line with compound heterozygous *CRB1* mutations (c.1892A > G and c.2548G > A) [61]. To replicate the *RPE65*-associated LCA and *AlPL1*-LCA phenotype in a 3D in vitro system, patient-specific iPSCs were generated that were later used to generate organoids [62,63]. A similar study explored pathogenic splicing variants of the *ABCA4* gene, a transporter protein responsible for Stargardt’s disease [64]. These studies have emphasized the importance of patient-specific, disease-targeted 3D model systems. A PR degeneration model was also generated using the NRL (neural retina leucine zipper) null phenotype, further mimicking S-cone syndrome and RP. This NRL-/- human-based 3D organoid system uncovered the possible role of MEF2C as a candidate regulator in cone development [65]. Since animal models restrict the understanding of the pathophysiology of X-linked RP (XLRP), human 3D retinal organoids provide an efficient, stable, and reproducible model system for exploring *RP2* mutation. As a proof of concept, a study was conducted to rescue the *XLRP* phenotype using AAV-mediated gene augmentation, preserving PRs [66].

### 2.2. Model for Validation of New Treatment Strategies

Furthermore, the protective effects of ophthalmic supplements such as 4-hydroxytamoxifen and diethylstilbestrol which were already established in retinal explants [67], were tested in in vitro 3D model systems [68]. 3D retinal organoids were generated to replicate general PR degeneration and used as testbeds to confirm their protective effects on the overall health of PRs. Recently, a protocol for retinal organoid generation underlined the essential conditions for optimal PR development. Growth factors, small molecules, and cell seeding density seem to significantly affect the numerical and functional efficiency in generating light-sensitive PRs. For the first time, these organoids were used as testbeds for evaluating the pharmacological effects of moxifloxacin (a retinotoxic agent at higher doses), resulting in successful replication of in vivo-like retinal cell damage that included loss of PRs and amacrine cells [49].

Retinal organoids are also helpful for exploring gene therapies for several retinopathies, for example by studying the gene delivery in a human system. One such study conducted recently indicates that AAV2-7m8 has superior transduction, possibly due to higher infectiousness and effective activation of secondary receptors [69]. Most genome editing techniques rely on highly specific endonucleases and the capacity of a cell to repair double-stranded breaks (DSB): since DSBs are cell-cycle dependent, it is even more important to examine the present gene editing options in vitro before starting clinical trials [70].

Modeling late-onset degenerative diseases could be difficult using stem cell-derived retinal organoids: stem cells rejuvenate during the reprogramming step and aging-associated epigenetic signatures are therefore lost [71]. It was previously thought that age-dependent aspects of diseases could not be replicated in human retinal organoids. However, a side-by-side comparison of retinal organoids with *post-mortem* human retinas based on single-cell transcriptomics has demonstrated that the different cell types within the organoids reach stable cellular states converging towards the ones from adult peripheral retinal cell types [14].

### 2.3. Retinal Organoids as Cell Sources for Therapeutic Transplantation

Photoreceptors are the vision-forming light-sensitive sensory neurons of the retina: their loss causes blindness. Since the human retina is unable to regenerate PRs or RPE intrinsically, the transplantation of donor cells has been intensively explored as a therapeutic option (Figure 1). Huge efforts have been made to generate human PR and RPE cells which can be successfully transplanted. Pluripotent stem cell-derived RPE cell replacement therapies have entered clinical trials for the treatment of diseases such as AMD and Stargardt’s disease [72,73,74,75]. In addition to RPE transplantation, there has also been increasing research efforts to develop strategies and techniques for restoring visual function by transplanting PRs [20,73,74,75,76,77,78,79,80,81,82,83]. However, the biomedical application of PRs still requires extensive research. The biggest problem for experimental PR transplantation approaches is finding a high-quality and quantity source of human PRs and demonstrating robust and functional PR integration into the host retina. Photoreceptors derived from retinal organoids have great potential for therapeutic photoreceptor transplantation [84,85]. Stem cell-derived photoreceptors can be further engineered, for example, by expressing optogenetic tools for studying functional integration into the host retina [86]. Obtaining sufficient quantities of transplantable cells is, however, still a challenge, despite the development of several retinal organoid protocols. As an alternative, controversial evidence has been presented suggesting that cytoplasmic transfer from grafted PR cells to remaining host PR cells can be beneficial, instead of physical and functional graft integration [86,87,88]. These findings require additional research, which will involve retinal organoids.

## 3. Conclusions

Retinal organoids are complex in structure and cellular organization. Therefore, it is challenging to recapitulate all morphological and functional features in vitro. Nevertheless, research in the last few years has revolutionized the field. From the pioneering work [28], several groups have focused on generating simple, efficient, less labor intensive, and less time-consuming protocols across various cell lines and cell types. Most importantly, new protocols must be reproducible. Testing combinations of several innovative approaches has made the generation of highly efficient and stable organoids easier. In summary, the relatively new field of human retinal organoids is dynamic and still growing. Several technical roadblocks and difficulties have already been overcome. Many research teams have significantly contributed to the detail-oriented, ongoing refinement of protocols for the generation of retinal organoids [89]. As a result, light-sensitive and functionally stratified human retinal organoids are now available. These are useful tools for basic research and disease modeling with ability for high throughput pharmaceutical drug screening and are complementary to the use of animal models. In addition, retinal organoids are being researched as a source of photoreceptors for cell transplantation therapy.

## Figures and Tables

**Figure 1 ijms-21-08484-f001:**
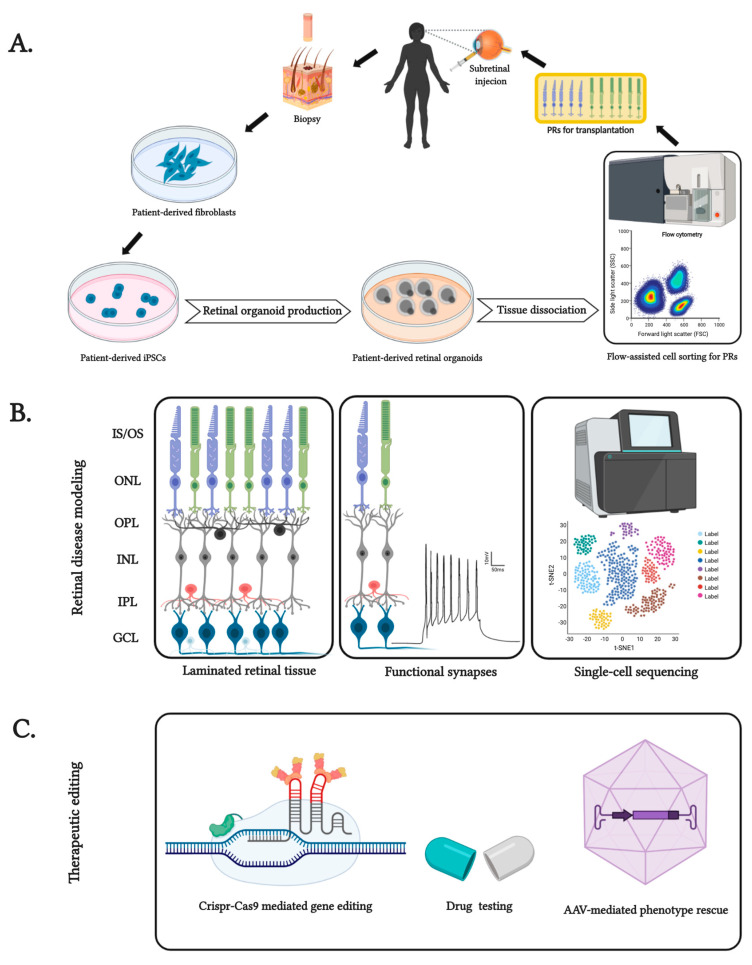
Features and applications of patient-specific induced pluripotent stem cell (iPSC)-derived retinal organoids. (**A**) Generation of patient genotype-specific autologous PRs extracted from mature retinal organoids. (**B**) Features of patient-derived retinal organoids, from its laminated cytoarchitecture containing inner and outer segments (IS and OS), inner and outer plexiform layer (IPL and OPL), inner and outer nuclear layer (INL and ONL), ganglion cell layer (GCL) and functionally recapitulated form to well-distributed cell types of the human retina for patient-specific disease modeling. (**C**) Use of retinal organoids as test beds for pharmaceutical interventions and genome editing techniques such as Crispr-Cas9-mediated and viral vector-mediated rescue of blindness phenotype for inherited retinal degeneration. Created with BioRender.com.

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
