# Peer review of "The Rise of Retinal Organoids for Vision Research"

_ijms, 2020, doi:10.3390/ijms21228484_

Round 1

Reviewer 1 Report

Review of IJMS-968295

Sharma, Krohne and Busskamp present a nicely-written, albeit brief review of the increase in the use of stem cell-derived retinal organoids as tool to investigate inherited eye disease. There are a few comments that should be addressed before this review is ready for publication.

 Comments

  1. In line 43 the authors state “with as many as 80 genes responsible”. This number is way off as some reports cite the number of different RP genes as many as 160+. Please amend and add appropriate, more up-to-date references.
  2. Line 64: please change “in-vitro” to “in vitro”. Also make sure this is consistent throughout the manuscript (also see lines 217 and 224 for further examples).
  3. Please thoroughly review the manuscript and make sure that ALL gene symbols/names are appropriately italicized. For example, in Line 176, “CEP290” should be “CEP290”.
  4. This is the most substantive comment! While the figure is nicely conceptualized and aesthetically-pleasing, with the lone exception being the suggestion of “intravitreal injection”. There is very little expectation that iPSC-derived photoreceptors will be delivered intravitreally. These cells will definitely need to be delivered to the subretinal space. Please amend and add appropriate references supporting subretinal vs intravitreal placement of PRs.
  5. Why does each reference in the bibliography include all authors except for reference #21? Please amend reference #21 to include all authors for consistency.

Author Response

We would like to thank the Reviewer for the nice word and feedback.  We have addressed all points and interspersed our responses.

Comments

1. In line 43 the authors state “with as many as 80 genes responsible”. This number is way off as some reports cite the number of different RP genes as many as 160+. Please amend and add appropriate, more up-to-date references.

Here we would like to respectfully disagree: we refer to the number of RP-causing genes, which is set to be >70 genes according to Retnet (https://sph.uth.edu/retnet/sum-dis.htm). Please see also the reference we are citing. We edited the text “…with over 70 genes…”. For streamlining the text, we did not go into details for individual mutations, of which more than 200 have been already identified.

2. Line 64: please change “in-vitro” to “in vitro”. Also make sure this is consistent throughout the manuscript (also see lines 217 and 224 for further examples).

We have consistently removed all hyphens.

3. Please thoroughly review the manuscript and make sure that ALL gene symbols/names are appropriately italicized. For example, in Line 176, “CEP290” should be “CEP290”.

We would like to thank the Reviewer for pointing this out. We have revised the manuscript accordingly.

4. This is the most substantive comment! While the figure is nicely conceptualized and aesthetically-pleasing, with the lone exception being the suggestion of “intravitreal injection”. There is very little expectation that iPSC-derived photoreceptors will be delivered intravitreally. These cells will definitely need to be delivered to the subretinal space. Please amend and add appropriate references supporting subretinal vs intravitreal placement of PRs.

We would like to thank the Reviewer for pointing this out. We have corrected this slip of the pen. Of course, photoreceptors must be subretinally transplanted. We have revised the figure.

5. Why does each reference in the bibliography include all authors except for reference #21? Please amend reference #21 to include all authors for consistency.

This is a good catch. We experienced technical problems with the citation manager used. We rechecked all citation formats in the revised manuscript.

Reviewer 2 Report

Kritika Sharma et al. produced a very well-written review article focusing on “The rise of retinal organoids for vision research”. I consider the manuscript very interesting and innovative but, in the same time, I suggest several revisions needed to improve the completeness and the readability of the paper:

  • Even if the manuscript is sufficiently wide, I suggest the authors to add, in Introduction section, more recent papers talking of new interesting biochemical pathways and interactors in retinal degenerations, that could become target of organoids experimentation. Regarding these, I suggest to add the following references to manuscript PMID:28764803, PMID: 32290199, PMID: 32413970 and PMID: 32326576.
  • Chapter 3 is, probably, “Conclusions” and not “Discussion”.
  • Finally, manuscript requires English revisions and typos correction.

Author Response

We would like to thank the reviewer for the nice assessment of our work. We have addressed all points and interspersed our responses.

  • Even if the manuscript is sufficiently wide, I suggest the authors to add, in Introduction section, more recent papers talking of new interesting biochemical pathways and interactors in retinal degenerations, that could become target of organoids experimentation. Regarding these, I suggest to add the following references to manuscript PMID:28764803, PMID: 32290199, PMID: 32413970 and PMID: 32326576.

We would like to thank the Reviewer for raising this point. Our intention was to be as concise as possible and having a focus on retinal organoid technology and existing applications, not on particular biochemical RP pathways that might be tested in retinal organoids in the future. The number of mutations and disease targets is very high, underlining the potential of organoid research by testing those in human 3D tissues. Consistently, we cannot reflect all possible targets and therefore we decided not to feature particular mutations and pathways. This is of course no judgement of the scientific quality of the proposed publications.

  • Chapter 3 is, probably, “Conclusions” and not “Discussion”.

We have changed the heading to “Conclusions”.

  • Finally, manuscript requires English revisions and typos correction.

Prior submission, our manuscript was professionally edited for grammar and spelling. Taking the Reviewer’s recommendation in mind, we have further revised and edited the manuscript.

Reviewer 3 Report

Retinal degeneration is a serious cause of blindness globally. Even though animal and human models are established to study retinal degeneration for many years, they are having their own limitations.  Stem cell derived retinal organoids model is considered as a replacement for these age old experimental models.  The review article by Sharma et al describes the retinal organoids in detail and covered the literature thoroughly on their development in retinal modeling. The article is well written, no grammatical errors. What are the advantages of 3D retinoid model over retinal explants

Author Response

We would like to thank the Reviewer for this positive feedback. It seems that the Reviewer’s report was truncated and potential specific comments were not communicated to us.

We have extended following statement to the introduction to explain the current limitations of post-mortemretinal tissue “Most of this research is based on animal models as human post-mortem retinal tissue is extremely fragile [13], not expandable, shows donor and preparation-dependent batch effects and is often unavailable.” Both, retinal organoids and post-mortem retinal tissues have their advantages and more importantly, culturing techniques will further improve in the future that both models will be invaluable for vision research. However, most labs don’t have access to human donor material. The recent paper by Cowan et al. Cell 2020 includes a very nice side-by-side retinal organoid and primary human retina morphological study. This work is referred to in our review, see citation [14].